# The Effect of Repeated Restraint Stress on Neuroglobin-Oligodendrocytes Functions in the CA3 Hippocampal Area and Their Involvements in the Signaling Pathways of the Stress-Induced Anxiety

Vlad-Alexandru Toma [1,2], Bogdan Dume [1], Rareș Trâncă [1], Bogdan Sevastre [3,*], Lucian Barbu [1], Gabriela Adriana Filip [4], Ioana Roman [2] and Alexandra-Cristina Sevastre-Berghian [4]

1    Department of Molecular Biology and Biotechnology, Babeș-Bolyai University, 400347 Cluj-Napoca, Romania
2    Institute of Biological Research, Branch of NIRDBS Bucharest, 400015 Cluj-Napoca, Romania
3    Paraclinic/Clinic Department, Faculty of Veterinary Madicine, University of Agricultural Sciences and Veterinary Medicine, 400372 Cluj-Napoca, Romania
4    Faculty of Medicine, University of Medicine and Pharmacy "Iuliu Hațieganu", 400347 Cluj-Napoca, Romania
*    Correspondence: bogdan.sevastre@usamvcluj.ro

**Abstract:** The present work shows the biochemical and structural fundamentals for the stress induced anxiety and stress adjustment response of the CA3 hippocampus area. Adult male Wistar rats were repeatedly exposed to a 3 h day restraint stress, for either 3 or 6 days. The concentration of corticosterone and testosterone in the CA3 hippocampus area was divergent, while oxidative stress was progressively increased during the stress exposure. The mitochondrial lysis in the CA3 neurons confirmed the oxidative stress events. Immunohistochemical findings showed that oligodendrocytes (OCs) proliferation and neuroglobin (Ngb) expression were stimulated, whereas MeCP2 expression was decreased as a balance reaction in stress exposure under corticosterone signaling. Remarkably, ultrastructural changes such as mitochondrial lysis, endoplasmic reticulum swelling, and perivascular lysis with platelets adherence to endothelium in the CA3 area were seen in the 6th day of restraining. The anxiety-like behavior was noticed 6 days later after stress exposure. These results suggest that the duration of the exposure, but not the intensity of the stress, is the key factor in the stress-buffering function by the CA3 hippocampus area via up-regulation of the Ngb-OCs bionome. The imbalance of the Ngb-OCs communication may be involved in the development of CA3-dependent anxious behavior.

**Keywords:** neuroglobin; CA3 pyramidal neurons; stress; buffering reactions; anxiety

## 1. Introduction

As defined by Selye, stress is the non-specific response of the body to any demand; the word "stress" is defined, alongside a physiological event, as a basic concept that radically changed the way of scientific thinking in the last century. Stress, from molecular events to behavioral changes, contains multiple forms, such as nitro-oxidative stress, hydric stress, and thermic stress or, without exhausting the list, psychological stress, the pivotal element in stress biology and pathology [1]. In all of these conditions, the biological models express their ability to balance the free radicals (ROS/RNS) generation, the immune response modulation after steroids action or memory impairment [2]. The nervous system is extremely sensitive to oxidative damage since it is rich in oxidizable substrates and has a high oxygen exposure and low antioxidant capacity. Further, the localization of major antioxidant defense systems in glial cells rather than in neurons may cause the nerve cells to be more susceptible to oxidants present in the brain exposed to neuropsychological stress. However, the oxidative stress generated by repeated restraining is a pleomorphic parameter,

and the difference between "harmless" and "bad" stress is made by the stressor's frequency as well as the exposure time as was noticed in Table 1.

**Table 1.** Experimental designs of restraint stress based on physical restraining or corticosterone injection in rats.

| | | Time | Reported measurements |
|---|---|---|---|
| [3] | Restraining 6 h/day | 11 days | ↑ serum corticosterone, adrenaline ↑ hippocampal ROS, NO, MDA, ↓ TAC |
| [4] | Restraining 1 h/day | 40 days | ↑ hippocampal MDA, ROS, ↓ TAC |
| [5] | Corticosterone 10–40 μg/Kg | 14 days | ↓ hippocampal TAC, ↑ hippocampal MDA, & ROS |
| | Restraining 4 h/day | 21 days | ↓ hippocampal TAC, ↑ hippocampal MDA, ROS |

ROS—Reactive Oxygen Species, NO—nitric oxide, MDA—malondialdehyde, TAC—total antioxidant capacity.

Early life exposure to neuropsychological stress determined a specific behavioral pattern in pups, which revealed their epigenetic mechanisms [6,7]. As noticed, the prenatal stress exposure induced in pups hypercorticosteronemia, anxiety [8,9], and behavior vulnerabilities such as the tendency to bipolar attitude, depressive features, or affective disorders [10,11]. Some studies [12–14] revealed that the stressed status of the individuals helps reduce the brain-infracted area in stroke by increasing the concentration of anti-inflammatory molecules such as interleukin 6, 10 (IL-6, IL-10), antioxidants like catalase (CAT), and reduced glutathione and promote the stress induced gliocytes proliferation. However, some published reports [12,15,16] related to our present work showed the failed co-localization of neuroglobin (Ngb) and astrocytic marker GFAP (glial fibrillary acidic protein) in the human brain following an acute ischemic stroke. Hundahl et al. [17,18] noticed that stress-exposed individuals depicted large brain infarcted areas, prominent inflammatory reactions, increasing oxidative stress, and slow formation of gliosis with a general low resistance to hypoxia and ROS/RNS actions, without Ngb protection against oxidative lesions. In these clinical or experimental situations, the role of Ngb remained questionable, as noticed by some authors [19,20] who mentioned a reduced infarct size even in Ngb-null mice compared with the wild-type controls.

Previous studies demonstrated that the CA3 hippocampus field played a stress-balancing function by activating the perineuronal glyocytes in a time-dependent manner [21]. The CA3 region is an exclusive part of the hippocampus, which was morphologically and functionally described as a notable connectome of the hippocampus [22]. The studies about CA3 pyramidal neurons [23–25] were largely based in part on the unique functional specializations formed by the mossy fiber inputs from the dentate gyrus, and the extensive axon collaterals between CA3 pyramidal neurons, which create a highly interconnected and excitable network [26–28]. Molecular features of the CA3 pyramidal neurons were described with high dynamics related to the characteristics of the stressor, such as decreasing glucocorticoid receptors (GRs) as a feedback reaction to hypothalamic-pituitary-adrenal axis (HPA) resistance for stressors [26,27].

In the context of restraint stress, the CA3 hypoxia was described [28,29] with HIF1$\alpha$ (hpoxia-inducible factor 1$\alpha$) and glycolytic enzymes involvement [30]. Some studies noticed, as a peripheral remark, that an increasing Ngb expression was correlated to attenuated cognitive deficit [31] and described Ngb and other molecules (neurokinin-A, catalase) as potential blood markers for generalized anxiety disorders [32,33]. Ngb, as well

as cytoglobin, in the absence of ligands (e.g., oxygen), showed hexacoordination by distal histidine, which contrasted the pentaccordination hem geometry seen in deoxygenated hemoglobin and myoglobin. Ngb displayed a high-affinity oxygen binding of approx. 1 Torr, and occurred at µM concentrations in the neurons as well as endocrine tissues [34]. However, the clear physiological function of Ngb is still unknown, besides Ngb intervention in hypoxia/stroke reactions that were noticed as enhancers for Ngb expression. The involvement of Ngb in cancer progression was also questionable. This ectopic role of the Ngb was derived from the cytoglobin function noticed as a cancer progression protein that was over-expressed in different types of tumors (non-small cell lung cancer, esophageal cancer, or melanoma) [34,35]. Our study was performed to investigate the role of stress in Ngb expression as a pivotal protein in brain oxidative stress with expected involvements in anxiety, as was suggested by previous results and other studies [33,36–38]. Alternatively, we have investigated the dynamic of the Ngb expression related to a gene-expression regulator methyl-CpG-binding protein 2 (MeCP2) and the time-dependent activation of the Ngb and CNP+ oligodendrocytes ($2'$,$3'$-cyclic nucleotide $3'$-phosphodiesterase OCs) couple. Few data were reported about Ngb and glial cells. Ngb was expressed in neurons but also in astrocytes and microglia. An expression link between Ngb and oligodendrocytes has not been demonstrated yet. However, a correlation between OCs and Ngb was noticed in neurodegenerative disorders, autism-spectrum disorders, heavy metals toxicosis, stroke, or traumatic brain injuries [39–41]. The research setup was started in previous studies, where Ngb was noticed with different expression patterns [36] in the frontal cortex after long-term treatment with metformin. The expression of the globins may also be ambiguously influenced by the methylation process [42]. However, DNA methylation in the $5'$regions of the globins genes might play a direct role in the regulation of gene expression [43]. Moreover, studies revealed that methylation of the Ngb gene influenced the tissue-specific expression pattern of the protein [44,45]. Other studies [46,47] have noticed that MeCP2 was involved in cytoglobin expression and in neuronal plasticity control, but available consistent data about MeCP2 and Ngb relation were not shown yet. Furthermore, MeCP2-Ngb interaction in stress also lacked knowledge that required experimental data. Our assumption had considered that all the stress reactions described until now depended on the sampling moment.

After repeated stress exposure, the CA3 area had developed a series of molecular, cellular, and, finally, behavioral stress-buffered reactions, which balanced the relation between stressors and coping behavior in a time-dependent manner. These reactions were interpreted as CA3-related stress tuners by co-enhancing of Ngb-OCs binome.

## 2. Materials and Methods

### 2.1. Animals

Adult Wistar male rats (3-month-old) weighing 300–350 g were provided ad libitum access to standard rat chow and water. Animals were maintained in a light/temperature-controlled room with a light/dark cycle of 12/12 h under 22 °C constant temperature. Rats were housed 7 rats/cage (60 cm × 40 cm × 20 cm). Experimental design is depicted in Figure 1.

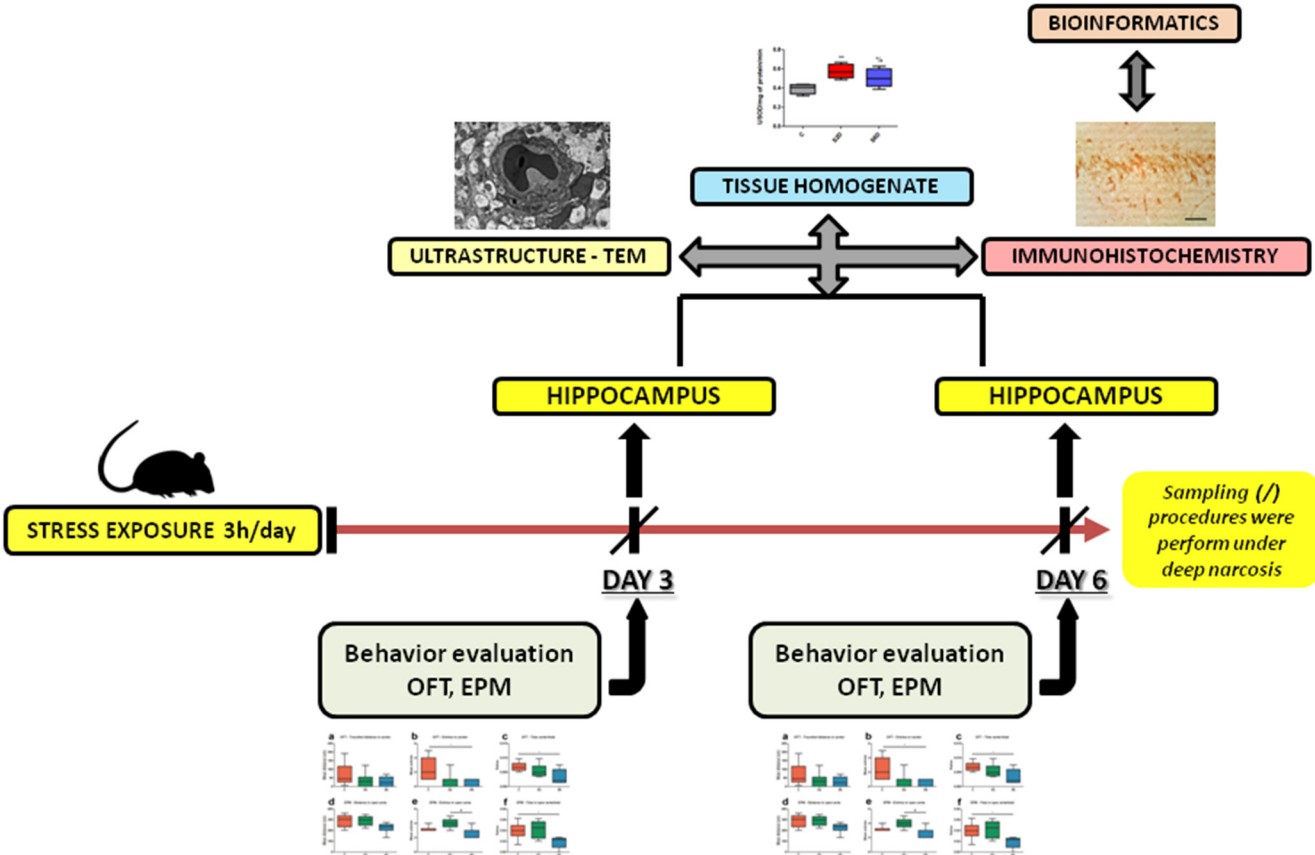

**Figure 1.** The experimental design that was applied in the current study.

## 2.2. Ethic Statement

Animal care and procedures were carried out in accordance with the Directive 2010/63/EU and national legislation. The project has been approved by the Ethical Committee of Babeş-Bolyai University (IRB no. 2012/3 February 2016). In the end, the animals were humanly killed by deep, prolonged, narcosis with isoflurane in an anesthesia cage. They were considered dead when no respiratory and heart activity was recorded. The irreversibility of the phenomena was assured by cervical dislocation.

## 2.3. Stress Induction

Restraint stress was induced by immobilization of the rats in plastic holders for 3 h/day [48]. The holders were plastic cylinders (7.6 cm × 20 cm, Kent Scientific, HLD-RL-T, Orringtonn, ME, USA) with an adjustable front nose cone for comfortable breathing. The rats were individually exposed (1 rat/holder) during 2 and 6 consecutive days to immobilization (3 h/day). Three experimental groups were used: Control (C), Restraint Stress for 2 days (S2) and Restraint Stress for 6 days (S6).

## 2.4. ELISA and Spectrohotometrical Assays

Dorsal hippocampus curvature (left CA3 area) was dissected and homogenized in a 1:1, mass/volume report with PBS-buffer saline 0.1 M pH 7.4, supplemented with protease inhibitor (cOmplete, 11697498001, Sigma-Aldrich, St. Louis, MO, USA). The lysates were centrifuged at 15,000 rpm for 30 min at 4 °C, and the final concentration of proteins was adjusted at 20 mg/mL (Bradford method). Using the supernatant, the levels of corticosterone (CS) and testosterone (TS) were assayed by ELISA methods (Abbexa Kits, Cambridge, UK). ELISA plates were then analyzed with a BioTek Synergy microplate reader (Biotek Instruments Inc., Winooski, VT, USA), according to the manufacturer's instructions. CAT activity was assayed by the kinetic method at 240 nm [18], whereas SOD

was assayed by the colorimetric method at 450 nm [49] using an UV-Vis spectrophotometer (VWR, UV-1600PC).

### 2.5. Behavior Tests

Two different tests were used in our study such as, OFT and EPM in order to assess the general locomotor activity and anxiety of the rodents, on the same groups of animals in the same day with 4 h between evaluations (OFT at 12 a.m., EPM at 16 p.m.). The animal's activity was quantified by a visual tracking system (Smart Basic Software version 3.0 Panlab Harvard Apparatus; Barcelona, Spain) using specific mazes for the rats (Ugo Basil Animal Mazes for Video-Tracking). Open Field Test (OFT): The animals were freely allowed to explore an open field arena (100 cm × 100 cm × 40 cm) for 5 min. The total travel led the distance and the total number of entered squares served as an index of general locomotor activity. Increases in central locomotion (number of entries and travelled distance in the center) or in time spent in the central part of the device (time spent in the center/total time) can be considered as anxiolytic-like behavior [50–52]. Elevated Plus Maze Test (EPM): The plus-shaped maze consists of two open (10 cm × 50 cm) and two closed (10 cm × 50 cm × 40 cm) arms that are 60 cm elevated above the ground level. Although, EPM is considered the gold standard for the evaluation of anxiety in basic research, it also measures motor activity. High open arms travelled distance, open arms number of entries and time ratio (open arms/total time) are considered relevant parameters of low anxiety-like behavior [50–52], whereas, total and closed arms travelled distance, total and closed arms entries are seen as an index of general locomotion in EPM. The animals were freely allowed to explore the maze for 5 min. Between tasks, the mazes were cleaned with 70% ethanol to remove any residual odor. A behavior evaluation was then performed 2 h before the animals were sacrificed.

### 2.6. Immunohistochemistry

For the evaluation of immunohistochemical features (Cell Signaling Technology Protocol), the brains were isolated and whole fixed in a 5% buffered neutral formalin solution for 48 h. After paraffin embedding, coronal sections were cut at 5 µm and mounted on Star Frost glass slides. The immunohistochemical stains were performed in brain tissues from all groups simultaneously to avoid differences in day-to-day experimental conditions. Tissue sections were dewaxed in xylene, and the rehydrated sections were subjected to HIER with citrate buffer pH 6 (Dako, S1699) in a steamer for 15 min. The slides were then treated with 3% hydrogen peroxide for 10 min. After washing with TBS, nonspecific background staining was blocked with 10% BSA in TBS pH 7.8 for 2 h. The sections were then incubated at 4°C for 12 h with primary antibodies (Sigma-Aldrich, St. Louis, MO, USA): rabbit anti-rat CNP (1:200, SAB5700668), mouse anti-rat MeCP2 (1:250, M6818), and rabbit anti-rat Ngb (1:850, N7162). The sections were then washed with TBS and treated with biotinylated-HRP link universal immunoglobulins in PBS (Dako, LSAB+ System-HRP, K0609) at room temperature for 30 min, incubated with streptavidin—peroxidase (Dako, LSAB+ System-HRP) for 30 min, and were treated with DAB (Dako, LSAB+ System-HRP) for 5 min. The slides were dehydrated and covered with synthetic resin (Titolchimica, TC 36259). Next, the slides were blind investigated by a histologist using an Optika trinocular microscope B383-FL with an MDC CCD Camera 2 MP. After microphotography, the images were prepared in Adobe Photoshop CS6 software to generate a whole figure. The number of positive cells was determined by the cell counter plug-in using Image J software.

### 2.7. Bioinformatics

For the investigation on the network of MeCP2 interactors, we used the STRING platform v11.0 [53] (https://string-db.org/, accessed on 8 March 2021). A basic search was performed for the MeCP2 protein, after selecting *Rattus norvegicus* as the targeted organism. To obtain even more insights, we opted for the extended network format, without clustering, since this feature would have been irrelevant for our purpose. Gene

Transcription Regulation Database (GTRD) platform v20.06 (http://gtrd.biouml.org/, accessed on 8 March 2021) was used for seeking possible implications of the MeCP2 in the transcription of Ngb. Both the EnsemblRat91 database manual query and the advanced feature of binding sites near the specified gene were used. The automatic search feature on the TRRUST v 2.0 (https://www.grnpedia.org/trrust/, accessed on 8 March 2021) was used for investigating a possible role played by MeCP2 in the Ngb regulation. To ensure that information was as recent as possible, we also downloaded all the entries for both human and mice (the only organisms present on this database) in TSV format and searched manually for MeCP2 influences on other genes. At first, Harmonizome (http://amp.pharm.mssm.edu/Harmonizome/, accessed on 8 March 2021) revealed to us the mechanisms through which MeCP2 protein regulated the expression of other genes. Then, we performed a manual search through all the results derived from the 89 datasets given by the platform. Even if MecP2 is not a veritable transcription factor, we wanted to assure that no important interactions were to be found on the TRANSFAC Public Database (http://gene-regulation.com, accessed on 8 March 2021), thus a search for the MeCP2 gene was performed in the appropriate section. For IntAct (https://www.ebi.ac.uk/intact/, accessed on 8 March 2021) and InterPro (https://www.ebi.ac.uk/interpro/, accessed on 8 March 2021) basic searches were performed. The feature of gene expression from the Ensembl Database (http://www.ensembl.org, accessed on 8 March 2021) was used to localize and assess the levels of expression for both MeCP2 and Ngb. Any format conversions were performed using the ID mapping tool hosted by UniProt (https://www.uniprot.org/uploadlists/, accessed on 8 March 2021).

### 2.8. Electron Microscopy (EM)

The hippocampal areas (oriented to CA3) after being immediately surgically removed, were chopped into 0.5 mm$^3$ pieces in an ice-cold buffered saline, which were then fixed in 2.5% phosphate 0.15 M phosphate-buffered glutaraldehyde for 90 min, washed in 20 vol. for three consecutive washes of 0.15 M phosphate buffer, then postfixed with 1% osmium tetroxide solution in 0.1 M phosphate buffer for 1 h and the tissue embedded in Epon 812 resin and sectioned at 50 nm thickness, using the Leica UC 6 ultramicrotome. Sections were transferred to coated specimen transmission EM grids, and after being double contrasted with uranyl acetate and lead citrate, they were examined in a JEOL JEM 1010 transmission electron microscope [54]. After microphotography, the EM images were clustered in composed figures using Adobe Photoshop CS6 software. The structures were identified based on the *Fine Structure of the Nervous System* [55] for ultrastructural nomenclature and our EM interpretations were compared to the work coordinated by Kurtz (1964) who described in detail the ultrastructural features of the cerebral tissue [56]. To avoid a qualitative analysis, the hippocampal samples were collected from all 7 animals from ipsilateral left CA3 areas of the hippocampus and the representative EM images were used to generate the whole figures in Adobe Photoshop CS6 software.

### 2.9. Statistics

The results are presented as box-plots, where the bottom and the top of the box are the first and third quartiles, respectively, and the whiskers above and below the box indicate the 95th and 5th percentiles. The median is indicated as a horizontal line. Biochemical and behavioral data were subjected to one-way analysis of variance (ANOVA) followed by Tukey's post hoc test when comparing all the groups. The Shapiro-Wilks test was used to test the normal distribution of the data. The scores for immunoreactions intensities of brain sections for each marker were analyzed by using rank-based nonparametric Kruskal-Wallis test with Dunn's test for multiple comparisons. In the ANOVA test, $p < 0.05$ was considered statistically significant. Tukey's Multiple Comparison Test and Dunn's test was considered statistically significant at $p < 0.05$ and was interpreted as follows: * $p < 0.05$, ** $p < 0.01$, *** $p < 0.001$ when comparisons were made with C group. Significant differences after comparisons S2 and S6 groups were noted as follow: # $p < 0.05$, ## $p < 0.01$, ### $p < 0.001$.

For each analysis, N (no. of rats or samples) was seven. Statistical analyses were completed using Graph Pad Prism version 5.0 for Windows, Graph Pad Software, San Diego, CA, USA.

## 3. Results

### 3.1. Restraint Stress Exposure Has Increased Oxidative Stress and Decreased the Testosterone (TS) Concentration in Hippocampus

The stress level, CS, oxidative stress, and TS were determined in the hippocampus (CA3 area). Stress exposure stimulated the HPA axis, which in turn, increased the concentration of CS (Figure 2a), as well as the oxidative stress (superoxide dismutase/SOD—Figure 1b, catalase/CAT—Figure 2c). The CS increase was a time-dependent reaction, thereby, repeated stress determined significant augmentation of CS in the hippocampus after 2 days ($p < 0.01$) and 6 days ($p < 0.001$) of restraining, as compared to Control. Additionally, after 6 days of repeated stress, CS concentration was twice as high as after 2 days of repeated stress exposure ($p < 0.01$). SOD increased after 2 days of stress ($p < 0.01$) as well as after 6 days of stress ($p < 0.05$), whereas CAT increased after 6 days of repeated stress exposure ($p < 0.001$). However, SOD activity followed a downward trend after 6 days of stress compared to 2 days of stress ($p < 0.05$). Alongside, TS concentration (Figure 2d) decreased after 2 days ($p < 0.01$) and 6 days ($p < 0.001$) of stress and this variation maintained the balance between glucocorticoids and sexual hormones.

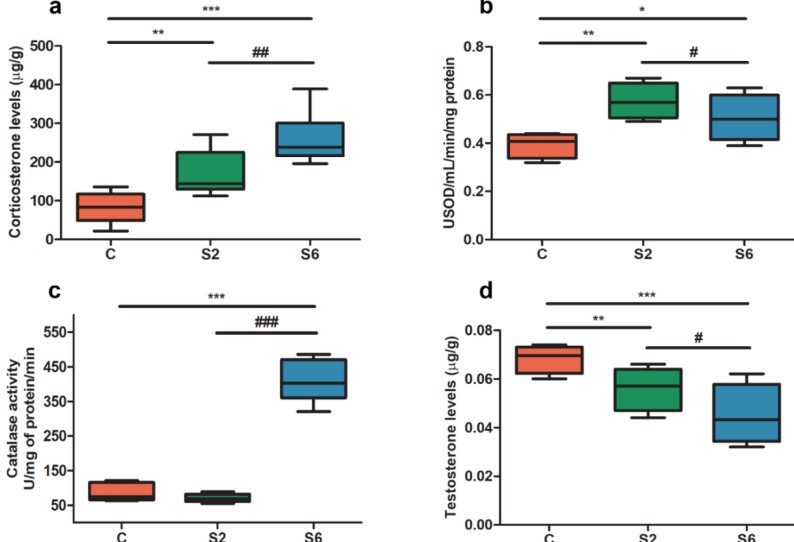

**Figure 2.** Repetitive restraint stress imbalanced the steroids levels and increased the activity of the antioxidant enzymes; (**a**) After repeated exposure to restraint stress, CA3 corticosterone (CS) levels were progressively increased ($p < 0.01$), whereas (**d**) testosterone (TS) concentrations were proportionally decreased after 2 days ($p < 0.01$) and 6 days ($p < 0.001$) of restraining; (**b**) SOD was increased after 2 ($p < 0.01$) and 6 ($p < 0.05$) days of stress, while (**c**) CAT was increased ~5 fold ($p < 0.001$) only after 6 days of repetitive stress exposure. The bottom and top of the box are the first and third quartiles and the whiskers above and below the box indicate the 95th and 5th percentiles. The median is indicated as a horizontal line. The experimental groups were distributed on the X axis. The Y axis was used to expose the measuring unit for the parameters. Statistical significance between S2 and S6 was noted with # (# $p < 0.05$; ## $p < 0.01$; ### $p < 0.001$), whereas the comparison to Control was annotated with * (* $p < 0.05$; ** $p < 0.01$; *** $p < 0.001$). F values: 2.75 (CS), 2.25 (TS), 9.05 (SOD), 86.24 (CAT).

### 3.2. Restraint Stress Has Induced an Anxiety-like Disorder

The effect of restraint stress on rats' locomotion tested in OFT and EPM was illustrated in Figure 3. Our results showed that restraint stress exerted no significant effects as compared to the control group on general locomotion in OFT and EPM tests ($p > 0.05$) (Figure 3a–h). The influence of restraint stress on the emotionality, tested in OFT and EPM, was exemplified in Figure 4. Regarding the locomotor activity and anxiety in OFT, the S6

group made fewer entries and spent less time in the center of open field arena as compared to C group (Figure 4b,c) (*p* < 0.05). Alongside, EPM has been successfully used to assess anxiety-like behavior in basic research, based on the natural tendency of rodents to explore novel environments. The animals typically spend a greater amount of time exploring the periphery of the arena or the closed arms of a maze, usually in contact with the walls, rather than the unprotected center area. Conversely, more time spent, higher travelled distance, and more entries made in the open arms of the EPM test apparatus during a 5 min test session can be considered as anxiolytic-like behavior [57]. In the EPM, the S6 group exhibited diminished number of entries compared to the S2 group (Figure 4e) (*p* < 0.05), and it spent less time in the open arms of the EPM apparatus as compared to the control (Figure 4f) (*p* < 0.05). Taken together, 6 days of restraint stress induced anxiety-like behavior based on EPM and OFT measurements.

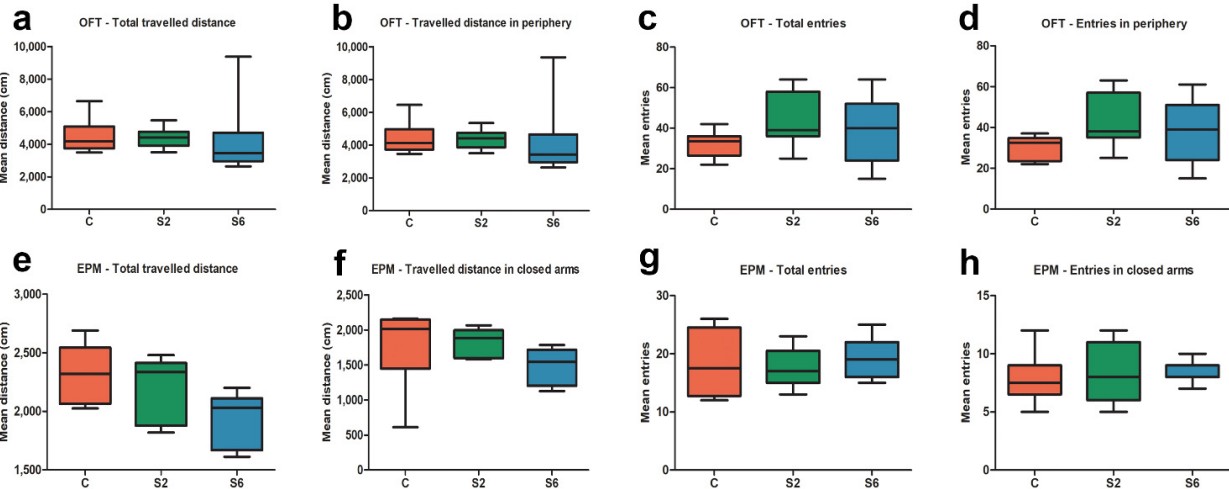

**Figure 3.** The effects of restraint stress on total (**a**,**e**) and peripheral (**b**,**f**) traveled distance and the total (**c**,**g**) and peripheral (**d**,**h**) number of entries in the open field test (OFT) and elevated plus maze (EPM) test. The restraint stress exerted no significant effects on general locomotion in OFT and EPM tests (*p* > 0.05) (**a**–**h**). Each group consisted of 10 rats. The bottom and top of the box are the first and third quartiles and the whiskers above and below the box indicate the 95th and 5th percentiles. The median is indicated as a horizontal line. The experimental groups were distributed on the X axis. The Y axis was used to expose the measuring unit for the test.

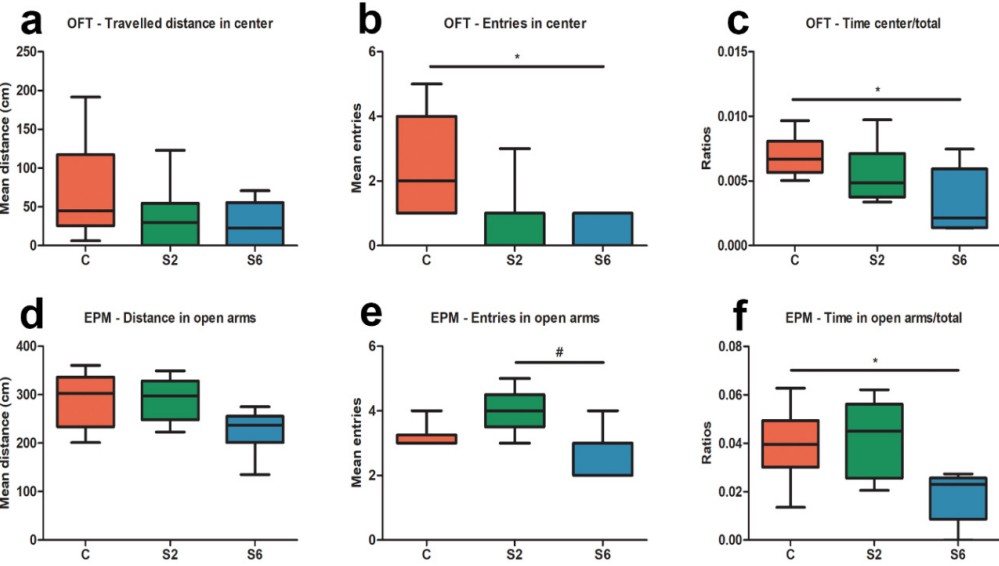

**Figure 4.** The effects of restraint stress on emotionality in the open field test (OFT) (**a**–**c**) and in the

elevated plus-maze (EPM) (**d**–**f**). The S2, and more prominently, the S6 rats exhibited a significantly lower number of entries and spent less time in the central part of the open field arena compared to the Control group (**b**,**c**) ($p < 0.05$). In the EPM test, the S6 rats made fewer entries in the open arms compared to S2 animals (**e**) ($p < 0.05$), whereas 6 days of restraint stress significantly decreased the time spent in the open arms compared to the Control group (**f**) ($p < 0.05$). Each group consisted of 7 rats. The bottom and top of the box are the first and third quartiles and the whiskers above and below the box indicate the 95th and 5th percentiles. The median is indicated as a horizontal line. The experimental groups were distributed on the X axis. The Y axis was used to expose the measuring unit for the test. The statistical significance between S2 and S6 was noted with # (# $p < 0.05$), whereas the comparison to Control was annotated with * (* $p < 0.05$).

### 3.3. The Stress Stimulated CNP⁺ OCs, Ngb, and MeCP2 Was Decreased, and MeCP2 Did Not Interact with Ngb Expression

Previous results mentioned that in repeated stress conditions, a type of glial cell is proliferated in the CA3 area [21]. The current data showed that CNP+ OCs were one of the gliocytes proliferated after different time of stress exposure (Figure 5a–d). Six days of stress induced a prominent OCs proliferation ($p < 0.001$) compared to Control, as well as to the S2 group. In order to verify if the expressions of Ngb and MeCP2 are convergent or divergent, we labeled these markers in the Control and experimental animals. Repeated stress induced, after 2 days ($p < 0.001$) and after 6 days ($p < 0.01$) exposure a decrease in MeCP2 expression (Figure 5e–h) in CA3 neurons. Conversely, the same stress period induced an increased expression of Ngb ($p < 0.001$) compared to the Control (Figure 5i–l). Additionally, compared to 2 days of stress exposure, 6 days of repeated stress increased Ngb expression ($p < 0.001$) in CA3 hippocampal area. However, the literature is relatively incomplete when it comes to Ngb regulation. Due to limited antibody availability, we investigated if there was any in silico link between the MeCP2 (Methyl-CpG-binding protein 2) (encoded by MeCP2) and the expression of Ngb gene, encoding neuroglobin. Using the STRING platform v.11.0 [53] we firstly thought to identify any basic interactions between the two proteins mentioned afore. The network had a total of 11 nodes and 36 edges, none of which revealed any relevant interaction with the expression of Ngb. Thus, at a first glance, only some indirect or distant pathways might be considered as viable connections between these two entities, since no direct interactions based on the amino acids sequence of MeCP2 were detected (Figure 6a). Searching in various databases (such as GTRD, TRRUST v 2.0-both human and mice, Harmonizome, TRANSFAC 7.0 Public Database, IntAct, and InterPro) led to no relevant information about interactions between MeCP2 and methylated Ngb or between their transcripts. This interesting inadequacy may come from the different levels of expression of both MeCP2 and Ngb in brain. Using the Ensembl Database, we observed that the highest single expression of Ngb in the brain was at 12 Transcripts Per Million (TPM) in documented rat experiments, while MeCP2 had the lowest expression at 17 TPM in brain, with values ranging from this number to 27 TPM in other experiments too (Figure 6b). The low expression levels of Ngb in the brain can be explained by the activity of TET1, which appeared to down-regulate the amount of Ngb transcripts by hypermethylation of its DNA sequence [58]. Considering this, it was quite clear that Ngb may have little or accidental interactions with MeCP2 due to Ngb not having a sustained expression in the cortex or hippocampus, and its regulation already being resolved plainly via TET1.

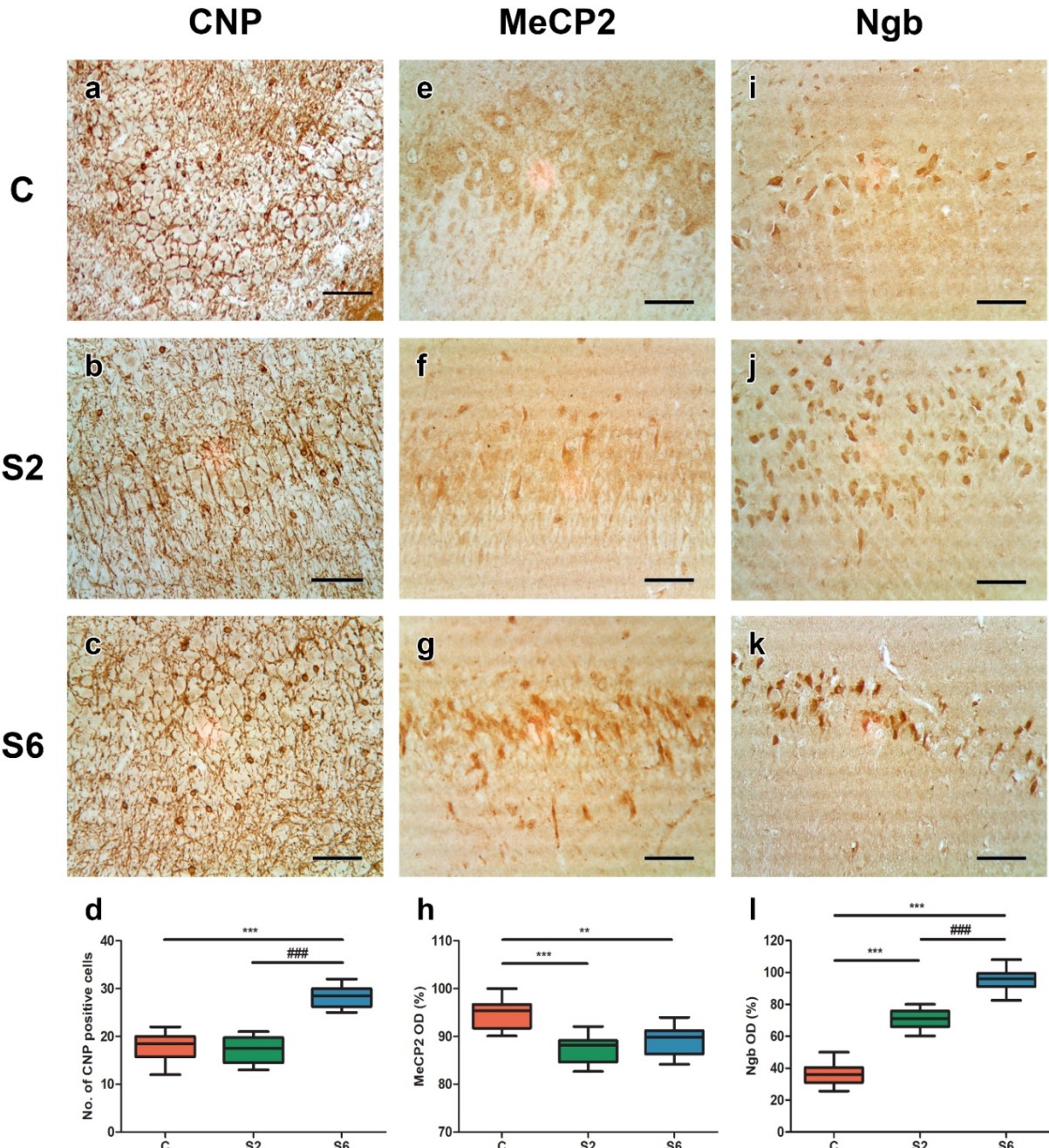

**Figure 5.** Immunohistochemical stain of rat hippocampus in the Control and stress-exposed animals; (**a–d**) Immunodetection of oligodendrocytes using CNP antibody (2′-3′-cyclic-nucleotide 3′-phoshodiesterase) revealed the significant oligodendrocytes proliferation after 6 days of repetitive stress ($p < 0.001$ compared to Control and S2 group); (**e–h**) MeCP2 immunoreactions decreased after repetitive stress exposure compared to Control ($p < 0.01$ for S6; $p < 0.001$ for S2); (**i–l**) Neuroglobin (Ngb) expression in CA3 neurons in the Control and stressed animals. The repetitive stress exposure significantly increased/stabilized Ngb expression compared to the Control ($p < 0.001$) or 2 vs. 6 days of stress ($p < 0.001$). The bottom and top of the box are the first and third quartiles and the whiskers above and below the box indicate the 95th and 5th percentiles. The median is indicated as a horizontal line. Statistical significance between S2 and S6 was noted with # (### $p < 0.001$), whereas the comparison to Control was annotated with (** $p < 0.01$; *** $p < 0.001$). The scale bar for (**a–k**) figures is 20 μm, magnification ×200.

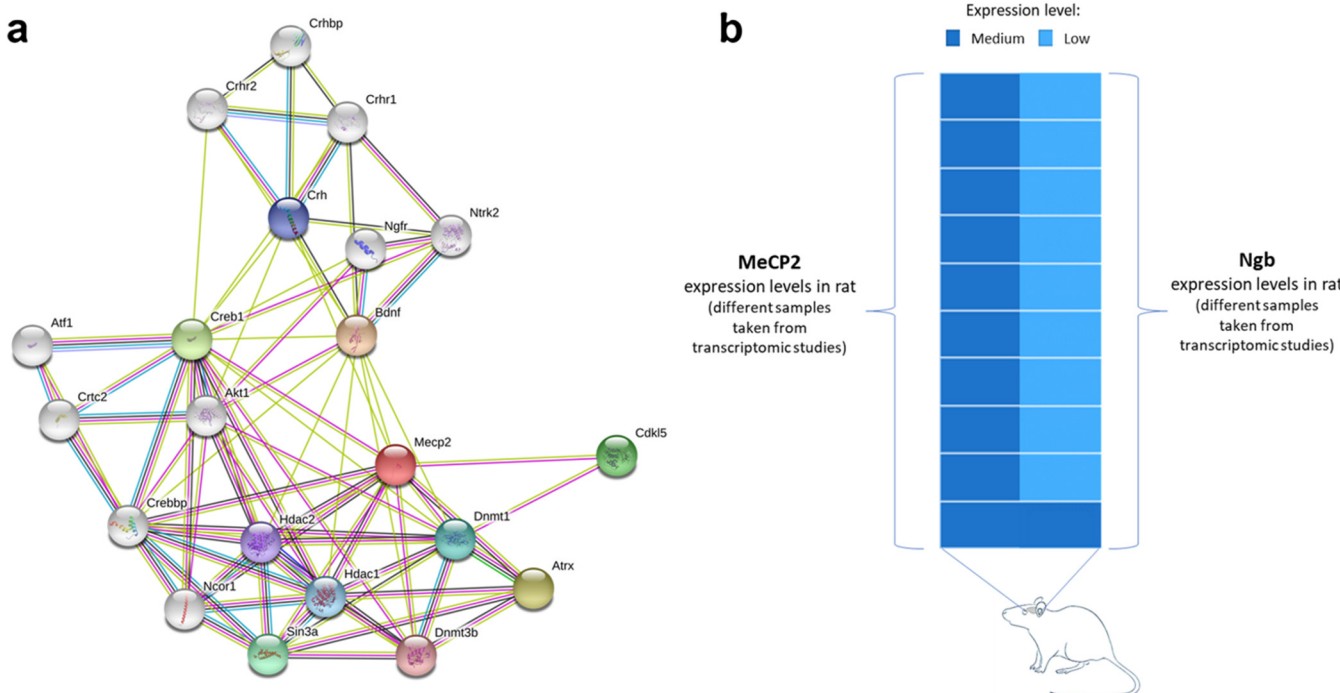

**Figure 6.** MeCP2 interactors' analysis. (**a**) The network of the MeCP2 interactors obtained by a queryin STRING v 11.0 revealed that no direct association between this protein and neuroglobin exists. Furthermore, the extended form of this matrix (shown in this figure) depicts that these two proteins do not share common interaction nodes (the important nodes have various colors, except grey) (**b**) A total of 10 different rat brain samples depict the gene expression contrast between MeCP2 and Ngb in rat cerebral tissue; in terms of specific expression levels, values of TPM > 10 were considered as 'Medium', with numbers ranging from 17 to27 TPM for MeCP2 expression and from 2 to 12 TPM for Ngb. Each tab represents a different sample analyzed in a transcriptomic study; these 10 sample sources were the only ones available at the time of the interrogation of Ensembl Gene Expression database).

### 3.4. Ultrastructural Pathological Changes Were Noticed in Mitochondrial-Vascular Structures

Ultrastructural investigations revealed a compact aspect without mitochondrial and endoplasmic reticulum (ER) swelling of the Control CA3 pyramidal neurons (Figure 7a,b) and rare irregular contours of the nuclei (Figure 7c) with slightly perivascular lysis (Figure 7d) after 2 days of repeated stress. Six days of repeated restraint stress induced prominent perivascular lysis (Figure 8a) associated with irregular contours of the nuclei (Figures 8b, 9b,c and 10d) and slight cytoplasm swelling (Figure 9c). Also, a platelet adherent to endothelium wall was noticed (Figure 9a), as well as dark neurons with dilation of the ER (Figures 8d and 10a,b,d). The mitochondrial integrity was varied between normal aspects with normal shapes and sizes and electron-dense curvilinear cristae, with some exhibiting trans-mitochondrial cristae coordination (Figure 9d) to total mitochondrial damage (Figure 10a,b). Repeated stress exposure for 6 days induced stromal aspect of the nucleus (Figure 8c) and proliferation of the oligodendrocytes (Figures 8c and 10c,d). The oligodendrocytes were characterized by nuclear chromatin blobs that were conspicuous and the narrow rim of cytoplasm contained much granular endoplasmic reticulum and numerous mitochondria.

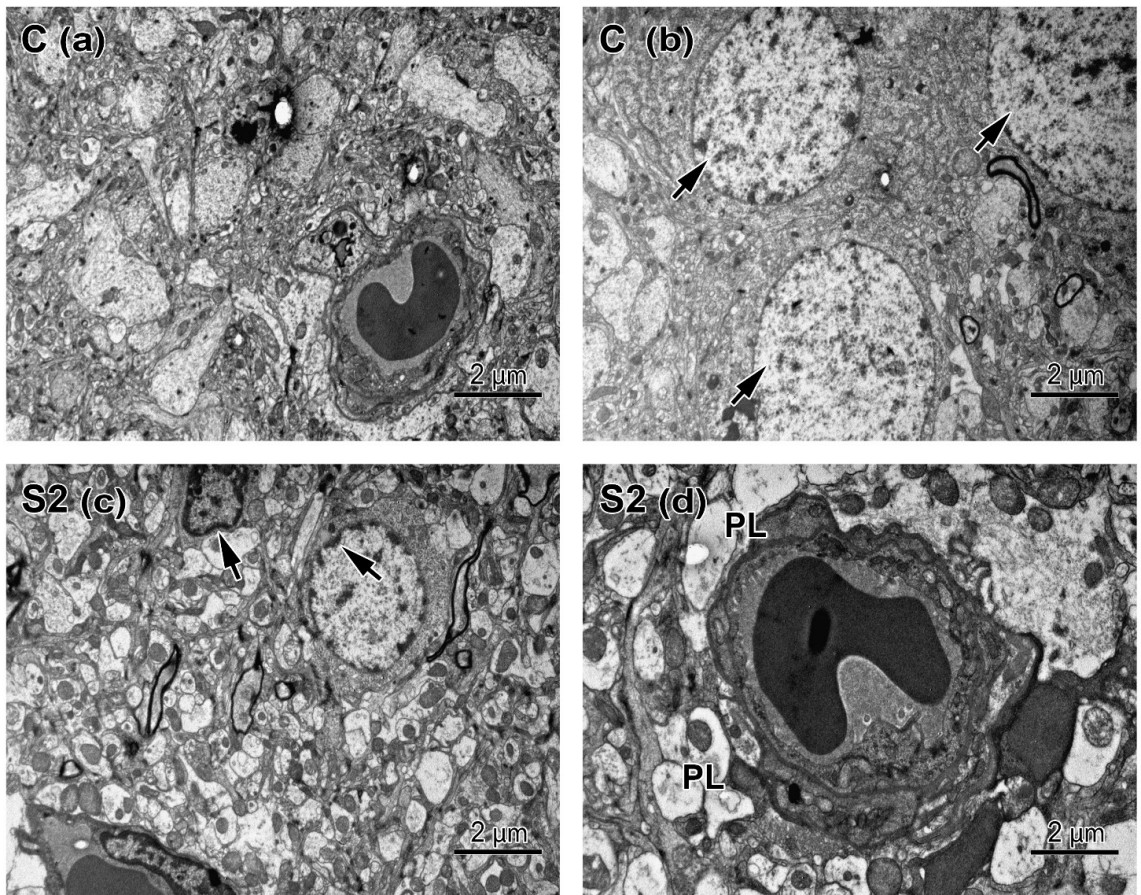

**Figure 7.** Electron micrographs showing neurons, glial cells, and capillaries. (**a**) The compact aspect of the neuropil in the CA3 area in Control; (**b**) The neuronal somas (black arrows) triad with regular nuclear shapes in the CA3 area of Control; (**c**) Oligodendrocyte and microglia without considerable changes in nuclear contours (black arrows) after 2 days of repeated restraint stress; (**d**) Perivascular lysis (PL) in CA3 hippocampal area after 2 days of repetitive exposure to restraint stress. Original magnification ×6000.

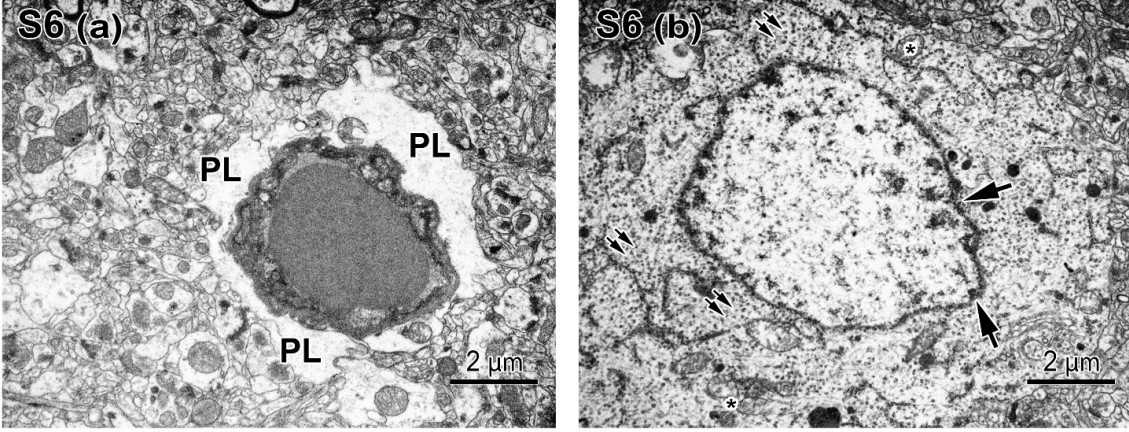

**Figure 8.** *Cont.*

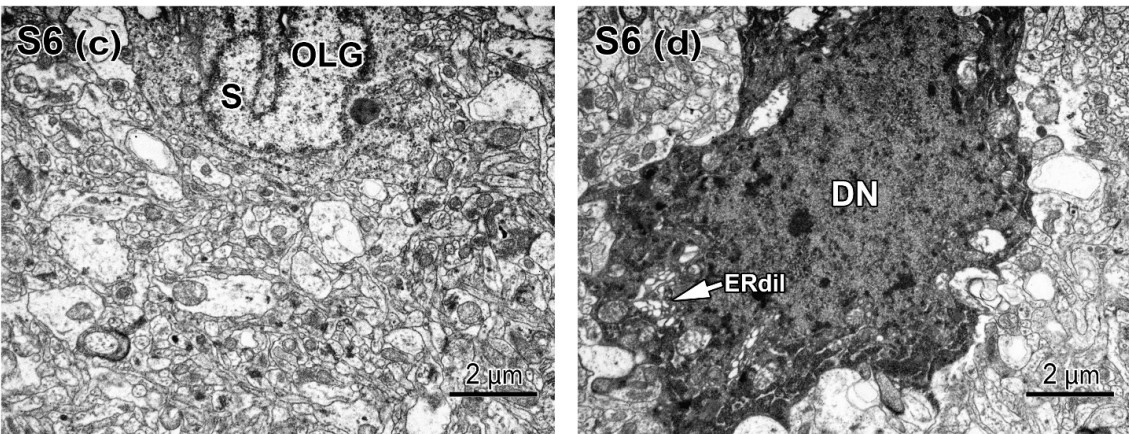

**Figure 8.** Electron micrographs showing neurons, oligodendrocytes, and vascular changes after 6 days of repetitive exposure to restraint stress; (**a**) Prominent perivascular lysis (PL) between CA3 neurons; (**b**) CA3 neuron with irregular nuclear shape (black arrows), mitochondrial lysis (asterisk), and frequent ribosomes (double arrows); (**c**) Oligonderocyte (OLG) with stromal aspect (S) of the nucleus; (**d**) Dark neuron (DN) with the dilated endoplasmic reticulum (ERdil, white arrow) in the CA3 hippocampal area; Original magnification ×6000.

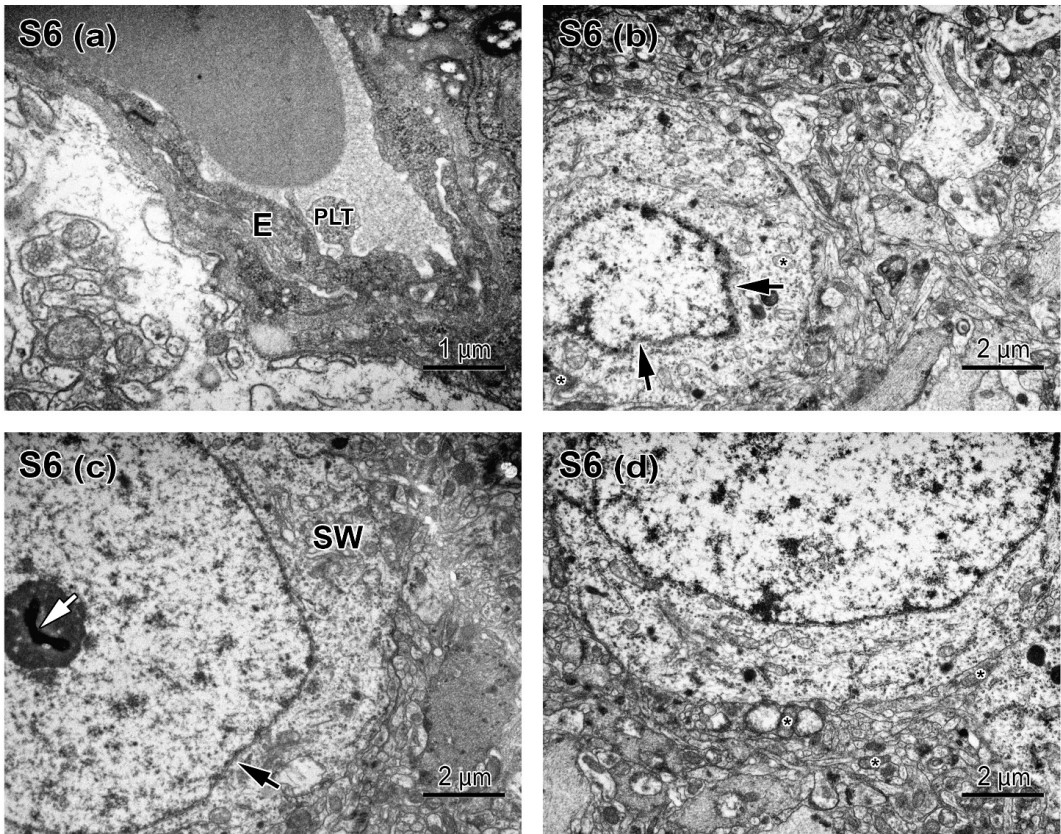

**Figure 9.** Electron micrographs showing the polyvalent changes in the CA3 area induced by 6 days of repeated restraint stress; (**a**) Platelet (PLT) adherent to endothelium wall (E) surrounded by perivascular lysis, which alters the transfer between astrocyte end-foot and capillaries; Original magnification ×12,000; (**b**) Neuron with irregular nuclear shape (black arrows) and near-normal mitochondrial aspect (asterisk); (**c**) CA3 neuron with slight irregularities in nuclear contour (black arrow) with cytoplasm swelling (SW) and condensed nucleolus chromatin (white arrow); (**d**) CA3 neuron with normal aspects of the mitochondria (asterisk) after 6 days of repeated stress; Original magnification ×6000.

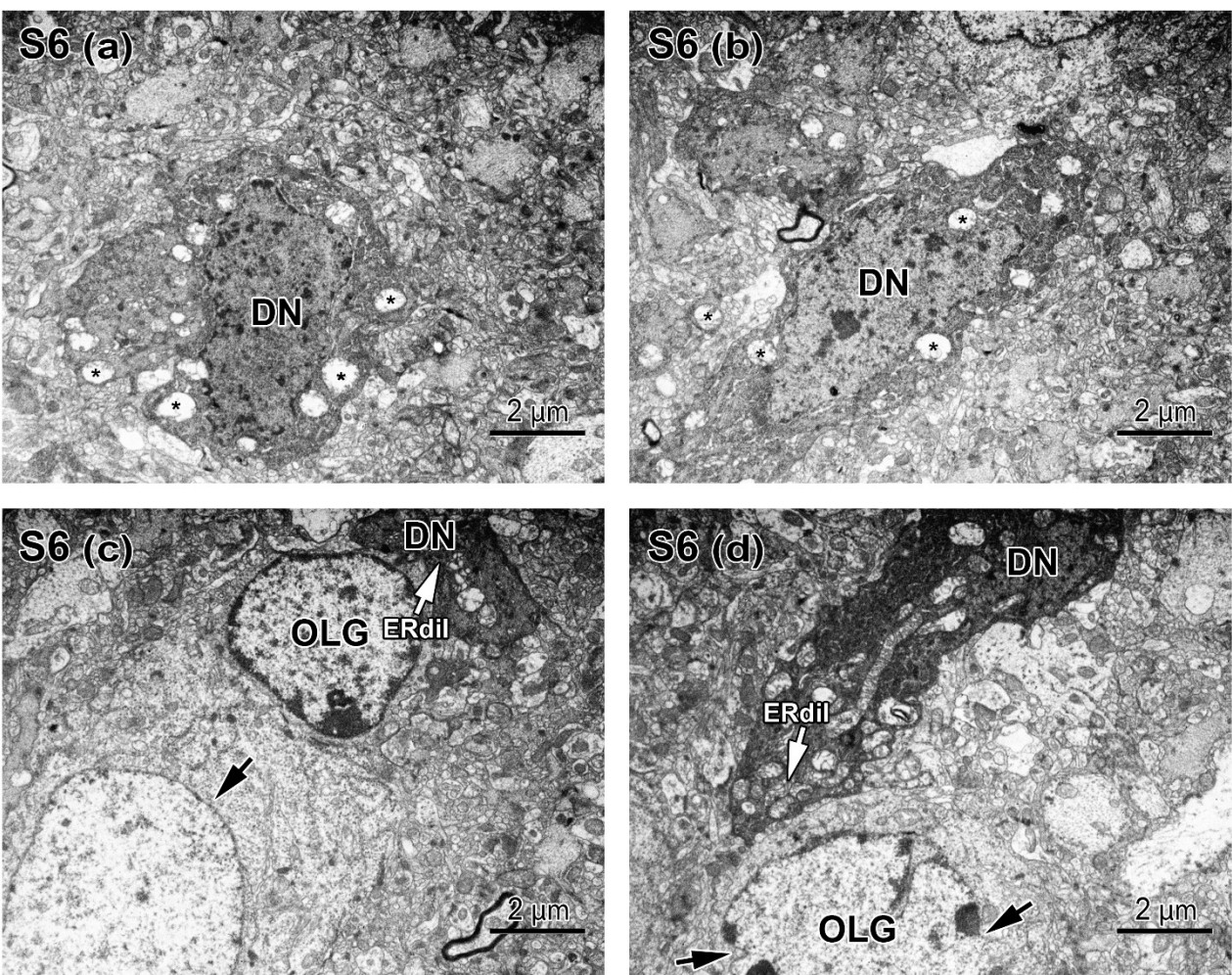

**Figure 10.** Electron micrographs of the CA3 neurons and oligodendrocytes after 6 days of repetitive exposure to restraint stress; (**a**) Small dark neuron (DN) with prominent mitochondrial lysis (asterisk); (**b**) Another aspect with one dark neuron and mitochondrial lysis (asterisk); (**c**) Dark neuron (DN) with dilated endoplasmic reticulum (ERdil, white arrow) in distal position to a neuron with normal aspect (black arrow). A satellite oligodendrocyte (OLG) marks the space between neurons; (**d**) Dark neuron (DN) with the dilated endoplasmic reticulum (ERdil, white arrow) with satellitosis of an oligodendrocyte (OLG) with indented nucleus (black arrows); Original magnification ×6000.

## 4. Discussion

### 4.1. Variability of the Oxidative Stress and Hormonal Response in Restraint Stress

Several studies were completed for prolonged periods of restraint stress exposure (between 10 and 40 days) and some of these studies were developed with intravenous corticosterone administration in rats. Their results indicated a substantial decline in antioxidant defenses by actions of corticosterone evidenced by coordinate decreases in the activities in the brain, liver, and heart of the free-radical scavenging enzymes. These results were significantly changed after corticosterone injection when the blood concentration was ~15–60 µg/dL, whereas endogenous hypercorticosteronemia was ~10–16 µg/dL according to our previous results, as well as with Czéh and Lucassen's available data [59]. Returning to our findings, short (2 days) and medium (6 days) time exposure to restraint stress for 3 h/day had gradually increased the corticosterone concentration, as well as SOD-CAT activity in the hippocampus and these data demonstrated that oxidative imbalance after restraining was started even after 2 days of repeated stress exposure. Son et al. [60] also mentioned that short-time exposure to psychiatric stress or physical stress was known to induce oxidative stress in the brain. Further, the same authors described the role of

sex hormones in stress coping reactions. Unlike estrogen, testosterone had not been well investigated in connection with its function on the brain exposed to restraint or psychological stress. Restraint stress significantly reduced testosterone levels in rats after 14 days of restraining (5 h/day). Our data were in accordance with other results and showed that even after 2 days of restraining, testosterone concentration was decreased in the hippocampus [60]. As mentioned by Lee et al., the hippocampal neurons and glial cells contained high levels of corticosterone receptors, which exposed them to glucocorticoids signaling during stress periods [3]. High corticosterone levels induced oxidative stress by stimulating glutamate-mediated ROS production and mitochondrial oxidative stress in neurons and glyocytes. Similarly to our results, several authors have reported that short or long-term periods of restraint stress decreased the blood concentration of testosterone and determined a prominent redox imbalance [61,62]. Based on both, in vitro and animal models, Handa et al. [63] and Bassil et al. [64] showed that corticosterone and HPA activation inhibited the testosterone production by the testes Leydig cells. The authors noticed that the absence of testosterone generated free radicals in male rats. Our findings also showed that oxidative stress enzymes and corticosterone were augmented whereas testosterone was decreased after 2 and 6 days of repeated stress [65]. Excessive corticosterone levels in the hippocampus accelerate neuronal inflammatory reactions by releasing pro-inflammatory cytokines such as IL-1$\beta$, IL-6, or TNF$\alpha$ [66].

### 4.2. Neuroglobin and Its Epigenetic Regulators in Repetitive Restraint Stress Exposure

The pleomorphic neuronal defense reaction to oxidative stress and pro-inflammatory cytokines induced neuroglobin (Ngb) expression [67]. Some studies described the up regulation of the Ngb in different pathological conditions such as traumatic brain injury [13] or stroke [14]. These data revealed that hemin specifically induced up regulation of Ngb in neurons. Moreover, Di Pietro et al. reported that mild traumatic brain injury determined a reversible increase in oxidative stress without Ngb stimulation, whereas severe traumatic brain injury caused increased Ngb expression [68]. Ragy et al. have noticed that in acute restraint stress (6 h, one day), Ngb expression was decreased and hemin intracerebral injection increased Ngb mRNA level. In the same study, the Ngb mRNA decrease was associated with low GSH concentration and high malondialdehyde level [14]. Our data demonstrated that Ngb immunoreactions in CA3 neurons were increased after repeated stress exposure. The apparent paradox with Ngb mismatch expression/immunoreactions was similar to HIF1$\alpha$ variations, in the oxygen sensing reactions, according to Ivan and Wu [69,70]. In the same way, Ngb may be stabilized or increased in hypoxic conditions, and mRNA could be decreased or increased, respectively, neuronal hypoxia being a major status in restraint stress [71]. Our data overlap with these findings. Thus, the testosterone decrease, oxidative stress enzymes, and corticosterone increase were strongly related to Ngb up regulation in CA3 neurons. Our assumption was initially that Ngb expression could be regulated by MeCP2, an epigenetic DNA methylation factor, involved in several neurodevelopment disorders like Rett syndrome, autism spectrum disorders, bipolar disorders, or cognitive deficit [72,73]. Our results have shown that MeCP2 immunoreactions were decreased after 2 and 6 days of repeated stress, in the same manner to Ngb increased. Apparently, there would be a link between Ngb and MeCP2 immunoreactions in CA3 neurons. We verified this observation using bioinformatics methods and the result was completely different: for Ngb regulation, a demethylation factor TET1 was involved and not MeCP2. However, further studies will probably demonstrate that MeCP2 is a reverse factor of TET1 regarding Ngb expression. Our data would suggest that the general gene expression related to MeCP2 repression was sustained during the restraint stress period.

### 4.3. Repeated Stress Increased the OCs Pool and Determined an Anxiety-like Behavior Based on New Reported Ultrastructural Changes

Besides CA3 neurons, the OCs were activated after repeated restraint stress. However, as we noticed in a previous study, the glial activation during stress exposure was a dynamic

process dependent on stress frequency [21]. Our data demonstrated that OCs proliferated in the CA3 area and around capillaries. The reliability of the panglial coupling in the CA3 area can be sustained as the proliferated OCs are not in direct contact with blood vessels and the metabolites reach neurons via astrocytes and OCs [74,75]. Based on this process, CA3 neurons were metabolically and mechanically protected by OCs and panglial coupling network was engaged in maintaining synaptic activity by metabolites delivery. Mitochondria critically determined the magnitude of hormonal stress responses and, conversely, mitochondrial functioning was expected to be closely related to mechanisms of stress regulation [76]. Our ultrastructural findings have shown that repetitive restraint stress induced abnormal angular features, matrix compartmentalization as well as markedly swollen mitochondria with peripherally placed, disorientated, and disintegrating cristae. As noticed by Fedoce et al. individuals with impaired mitochondria function would be vulnerable to stress-related depletion of the nervous tissue's energy resources and, hence, to the development of different psychopathologies [77]. Recently proposed, a mitochondrial etiology of neuropsychiatric disorders, linking mitochondrial dysfunction to depression, anxiety, schizophrenia, or bipolar disorders [78,79]. Our results revealed that 6 days of repeated restraint stress have induced an anxiety-like behavior, which would become a full anxious behavior if the stress period would continue. Our findings were supported by the definition of normative anxiety, which described that GSH and catalase balanced the effect of corticosterone. In turn, in high anxiety, the ROS generation after neurosteroids signaling was no more under antioxidant control as previously described [76,80]. On the assumption that repeated stress was balanced by the CA3 area (perineuronal OCs and neuronal changes), the anxiety-like behavior was a result of these gradual balancing mechanisms. As ultrastructural degradation in the neuron evolved, OCs activation increased together with Ngb expression as well as SOD-CAT activity. These events maintained the histological features of the CA3 area and had contributed to the delayed onset of anxiety after 6 days of repetitive exposure to restraint stress. Moreover, according to our findings, repeated stress has determined platelet adhesion to endothelial cells and promoted atherosclerosis, as well as the emotional-based triggering of acute coronary disease. As was described by Barbucci et al., high plasma corticosterone (~270 ng/mL) induced numerous pseudopodia with fused membranes in platelets. Freedman et al. correlated oxidative stress with thrombotic response starting with platelet adhesion to endothelium. Brydon et al. have demonstrated that repetitive stress impaired serotonin signaling, which induced the expression of PSGL-1 (P-selectin-glycoprotein ligand 1) in platelets and supported the platelets adhesion [80–82]. In light of relationships between serotonin signaling, oxidative stress, and PSGL-1 expression, the platelets adhesion to CA3 capillaries supported the anxiety behavior and suggested that after 6 days of repetitive action of the stressors, the polyfactorial eustress response was configured in distress pleiomorphic reactions. The balance between eustress and distress was established by the humoral response (antioxidant defense, neurosteroid signaling, Ngb expression) as well as with the oligodendrocyte's proliferation. Based on these balancing reactions, the CA3 neurons had kept their structural integrity in a time-dependent manner. As an overview, the stress was mild—it seemed that the frequency, not the intensity was the key element in the CA3 buffering reactions.

## 5. Conclusions

Neuroglobin expression plays a pivotal role in stress coping mechanisms due to its TET1-regulated oxygen-sensing role in the microenvironment of CA3 neurons. The presence of many oligodendrocytes between these neurons demonstrated that the CA3 area was a distinct reactive site of the hippocampus, involved in stress control and adaptation. Six days of repetitive restraint stress had induced CA3 plasticity by gradually increasing the number of oligodendrocytes and Ngb expression. Further, after 6 days of repetitive stress, the molecular and morphological changes were translated into anxious behavior.

**Author Contributions:** Conceptualization: V.-A.T, B.S. and A.-C.S.-B.; methodology: V.-A.T, B.S., A.-C.S.-B., B.D. and I.R.; software: R.T, V.-A.T, B.D. and A.-C.S.-B.; validation: V.-A.T, A.-C.S.-B., L.B., R.T. and G.A.F.; formal analysis: V.-A.T, I.R., R.T, A.-C.S.-B. and L.B.; investigation: V.-A.T, A.-C.S.-B. and L.B.; resources: V.-A.T, B.S. and A.-C.S.-B.; data curation: V.-A.T. and A.-C.S.-B.; writing—original draft preparation: V.-A.T. and A.-C.S.-B.; writing—review and editing: V.-A.T, B.S. and A.-C.S.-B.; visualization: V.-A.T, B.D., I.R., G.A.F. and B.S.; supervision: B.S. and G.A.F.; project administration: V.-A.T, B.S. and A.-C.S.-B.; funding acquisition: V.-A.T. and A.-C.S.-B. All authors have read and agreed to the published version of the manuscript.

**Funding:** The present work received financial support through the project: Entrepreneurship for innovation through doctoral and postdoctoral research, POCU/380/6/13/123886 co-financed by the European Social Fund, through the Operational Program for Human Capital 2014–2020, and by a grant of the Ministry of Research, Innovation and Digitization, CNCS—UEFISCDI, project number PN-III-P1-1.1-TE-2021-0159, within PNCDI III (TE60/2022).

**Institutional Review Board Statement:** The animal study protocol was approved by the Institutional Review Board (IRB) of Babes-Bolyai University, IRB no. 2012/3 February 2016.

**Informed Consent Statement:** Not applicable.

**Data Availability Statement:** Not applicable.

**Conflicts of Interest:** The authors declare no conflict of interest. The funders had no role in the design of the study; in the collection, analyses, or interpretation of data; in the writing of the manuscript, or in the decision to publish the results.

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
