# Peer review of "The Effect of Repeated Restraint Stress on Neuroglobin-Oligodendrocytes Functions in the CA3 Hippocampal Area and Their Involvements in the Signaling Pathways of the Stress-Induced Anxiety"

_applsci, doi:10.3390/app12178680_

Round 1

Reviewer 1 Report

The following article presents data on an important area related to the role of chronic stress on hippocampal function and involvement in signaling pathways involved in anxiety. Overall this is a well-written comprehensive study to evaluated the role of Neuroglobin (NGb), gene expression and cellular changes in hippocampus related to restraint stress.

Introduction:

The introduction provides a good “general” synopsis of stress mechanisms involvement in brain. However, there are several areas that can be improved for overall flow and understanding of how the authors got to the specific research questions related to “Ngb expression related to a gene-expression regulator MeCP2 and the time-dependent activation of the Ngb and CNP+ oligodendrocytes (OCs) couple.” MeCP2 was only mentioned in this sentence as a gene expression regulator, however much more can be emphasized to explain its importance for investigation in stress and hippocampus and neuropsychiatric disorders, as well that the previous work done that relate specifically to this study. For CNP+ OCs, the acronym was used “CNP” without any description or explanation. This issue with acronyms is present throughout the manuscript with other abbreviations, please check this. In addition to these comments for the introduction and related to the same discussion here, a better description of Ngb and its role in the nervous system is needed in introduction. Are there any other areas of research besides the hypoxia/stroke literature that depicts mechanisms of Ngb that could potentially be related to stress, anxiety or hippocampus?

Methods:

In ethical statement, more detail on agent used for euthanasia is necessary for statement and for understanding results related to stress in these animals.

Check abbreviations in this section (e.g., CAT activity).

Under behavioral tasks: good descriptions of the OFT and EPM, however it is not consistent between two. For EPM and OFT, the rationale for use is well explained, however what was actually measured for the study- this could be a simple re-wording. For EPM the actual duration of test was not stated. OFT should be described as locomotor activity and anxiety, not “emotionality” this should also be corrected throughout manuscript.

Sections 2.6-2.8 are nicely described.

In section 2.9 statistics: for behavioral tests were data subjected to test for normalcy and equal variance for the parametric ANOVA test? Based on figures 1 and 2, I would request that you state your statistical approach for choosing ANOVA based on these checks. Another discrepancy is with the animal numbers. Line 211: For each analysis, N (no. of rats or samples) was seven; figure legend #2- “Each group consisted of 10 rats.”.  Please clarify and correct if necessary.

Results:

Figure legends (1 & 2): need to label x-axis or define in legend. It would be helpful to see the scatter overlaid on the box-plots for these behavioral data.

Figure 4: image scaling on panels a-k not described.

Figure 5B should be represented/graph differently and explained better. It is unclear what the authors are trying to show with the panel with the different icons/hyperlinks. This appears to just be a “screenshot” from the database.

Discussion:

Table 1 and explanation of differences in stress may be more helpful to the reader in the introduction section with the discussion section focused on specific interpretation of the data from this study with discussion its stress condition specifically.

Author Response

Comments Reviewer #1

The following article presents data on an important area related to the role of chronic stress on hippocampal function and involvement in signaling pathways involved in anxiety. Overall this is a well-written comprehensive study to evaluated the role of Neuroglobin (NGb), gene expression and cellular changes in hippocampus related to restraint stress.

Introduction:

  1. The introduction provides a good “general” synopsis of stress mechanisms involvement in brain. However, there are several areas that can be improved for overall flow and understanding of how the authors got to the specific research questions related to “Ngb expression related to a gene-expression regulator MeCP2 and the time-dependent activation of the Ngb and CNP+ oligodendrocytes (OCs) couple.” MeCP2 was only mentioned in this sentence as a gene expression regulator, however much more can be emphasized to explain its importance for investigation in stress and hippocampus and neuropsychiatric disorders, as well that the previous work done that relate specifically to this study.

A: Thank you for your suggestion; the issue was resolved in Introduction (view comments according to Rev#1 in manuscript and the text is with blue color).

[Few data were reported about Ngb and glial cells. Ngb was noticed as a protein expressed beside neurons by astrocytes and microglia but an expression link between Ngb and oligodendrocytes is still not demonstrated. However, a correlation between OCs and Ngb was noticed in neurodegenerative disorders, autism-spectrum disorders, heavy metals toxicosis, stroke, or traumatic brain injuries [39-41]. The research setup was started in previous studies where Ngb was noticed with different expression patterns [36] in the frontal cortex after long-term treatment with metformin. Also, the methylation of the globins was reported with an ambiguous effect on the expression of the globins [42]. However, DNA methylation in the 5′regions of the globins genes might play a direct role in the regulation of gene expression [43]. Moreover, studies revealed that methylation of the Ngb gene influenced the tissue-specific expression pattern of the protein [44, 45]. Other studies [46, 47] have noticed that MeCP2 was involved in cytoglobin expression and in neuronal plasticity control, but available consistent data about MeCP2 and Ngb relation were not shown. Furthermore, MeCP2-Ngb interaction in stress was also a lack of knowledge that required experimental data.]

  1. For CNP+ OCs, the acronym was used “CNP” without any description or explanation. This issue with acronyms is present throughout the manuscript with other abbreviations, please check this.

A: During the manuscript, all acronyms were described (with red color). The issue was resolved.

  1. In addition to these comments for the introduction and related to the same discussion here, a better description of Ngb and its role in the nervous system is needed in introduction. Are there any other areas of research besides the hypoxia/stroke literature that depicts mechanisms of Ngb that could potentially be related to stress, anxiety or hippocampus?

A: Thank you for your suggestion; the issue was resolved in Introduction (view comments according to Rev#1 in manuscript and the text is with blue color).

[Ngb, as well as cytoglobin, in the absence of ligands (eg. oxygen), show hexacoordination by distal histidine which contrasts the pentaccordination hem geometry seen in deoxygenated hemoglobin and myoglobin. Ngb displays high-affinity oxygen binding of approx. 1 Torr and occurs at µM concentrations in the neurons as well as endocrine tissues [34]. However, the clear physiological function of Ngb is still unknown, besides the Ngb intervention in hypoxia/stroke reactions that were noticed as enhancers for Ngb expression. The involvement of Ngb in cancer progression was also questionable. This ectopic role of the Ngb was derived from cytoglobin function that was noticed as a cancer progression protein that was over-expressed in different types of tumors (non-small cell lung cancer, esophageal cancer, or melanoma) [34, 35]. ]

Methods:

  1. In ethical statement, more detail on agent used for euthanasia is necessary for statement and for understanding results related to stress in these animals.

A: At the end the animals were humanly killed by deep prolonged narcosis with isofluran in anestesia cage.

  1. Check abbreviations in this section (e.g., CAT activity).

A: issue solved

  1.  Under behavioral tasks: good descriptions of the OFT and EPM, however it is not consistent between two. For EPM and OFT, the rationale for use is well explained, however what was actually measured for the study- this could be a simple re-wording. For EPM the actual duration of test was not stated. OFT should be described as locomotor activity and anxiety, not “emotionality” this should also be corrected throughout manuscript.

A: Issue solved. The changes are with colored in manuscript, adressed to Rev#1.

Two different tests were used in our study, such as, OFT and EPM, to assess the general locomotor activity and anxiety of the rodents, on the same groups of animals in the same day, with 4 h between evaluations (OFT at 12 a.m, EPM at 16 p.m.).

The total travel led distance and the total number of entered squares served as an index of general locomotor activity. Increases in central locomotion (number of entries and travelled distance in the center) or in time spent in the central part of the device (time spent in the center/total time) can be considered as anxiolytic-like behavior.[50-52]. Elevated Plus Maze Test (EPM): The plus-shaped maze consists of two open (10×50 cm) and two closed (10×50×40 cm) arms that are 60 cm elevated above the ground level. Although, EPM is considered the gold standard for the evaluation of anxiety in basic research, it also measures motoractivity. High open arms travelled distance,open arms number of entries and time ratio (open arms/total time) are considered relevant parameters of low anxiety-like behavior [50-52], whereas, total and closed arms travelled distance, total and closed arms entries are seen as an index of general locomotion in EPM. The animals were freely allowed to explore the maze for 5 minutes. Between tasks,the mazes were cleaned with 70% ethanol to remove residual odor. Behavior evaluation was performed 2 hours before the animals were sacrificed.

Sections 2.6-2.8 are nicely described.

  1. In section 2.9 statistics: for behavioral tests were data subjected to test for normalcy and equal variance for the parametric ANOVA test? Based on figures 1 and 2, I would request that you state your statistical approach for choosing ANOVA based on these checks. Another discrepancy is with the animal numbers. Line 211: For each analysis, N (no. of rats or samples) was seven; figure legend #2- “Each group consisted of 10 rats.”.  Please clarify and correct if necessary.

A: The aspects were clarified. The correction of section 2.9. was marked with blue color.

The median is indicated as a horizontal line. Biochemical and behavioral data were subjected to one-way analysis of variance (ANOVA) followed by Tukey’s post hoc test when comparing all the groups. The Shapiro-Wilks test was used to test the normal distribution of the data. The scores for immunoreactions intensities of brain sections for each marker were analyzed

 Results:

  1. Figure legends (1 & 2): need to label x-axis or define in legend. It would be helpful to see the scatter overlaid on the box-plots for these behavioral data.

A: Issue solved.

  1. Figure 4: image scaling on panels a-k not described.

A: Issue solved.

  1.  Figure 5B should be represented/graph differently and explained better. It is unclear what the authors are trying to show with the panel with the different icons/hyperlinks. This appears to just be a “screenshot” from the database.

A: Issue solved. The track changes revealed the correction. Also, an improved Figure was attached to the manuscript.

 Discussion:

  1. Table 1 and explanation of differences in stress may be more helpful to the reader in the introduction section with the discussion section focused on specific interpretation of the data from this study with discussion its stress condition specifically.

A: Thank you! Issue solved.Bottom of Form

Reviewer 2 Report

In this form, the article cannot be accepted for publication.

Notes:

1) The English language needs to be seriously rewritten. I would recommend authors to show the work to a native speaker.

2) The figures are absolutely unreadable, which does not allow for a normal examination of the work. Figures must be submitted in high resolution.

3) In the materials and methods, a figure with an experimental scheme should be presented.

4) Headings in the results are excessively long. It needs to be reworded briefly.

5) The discussion is written redundantly. It is necessary to shorten and pay attention to the discussion of the results obtained.

Author Response

Comments Review #2

1) The English language needs to be seriously rewritten. I would recommend authors to show the work to a native speaker.

A: The manuscript was carefully check by a native English speaker with expertise in medical research.

2) The figures are absolutely unreadable, which does not allow for a normal examination of the work. Figures must be submitted in high resolution.

A: All Figures all now available as separate files in high resolution. Issue solved.

3) In the materials and methods, a figure with an experimental scheme should be presented.

A: Fig. 1 depicts the experimental design. Issue solved.

4) Headings in the results are excessively long. It needs to be reworded briefly.

A: All headings of the Results section were shortened.

5) The discussion is written redundantly. It is necessary to shorten and pay attention to the discussion of the results obtained.

A: This section was shortened by ~ 1 page, so we consider that the issue was solved. Thank you for all your suggestions.

Reviewer 3 Report

The paper describes the in vivo and in vitro testing performed to investigate the role of stress in Ngb expression as a pivotal protein in brain oxygen sensing reactions with expected involvements in anxiety progression. Of particular of interest is the main conclusion of the manuscript - - it seems that the frequency, not the intensity is the key element in the CA3 buffering reactions. Overall this is a well designed study and provides very interesting results. With some minor editing to the manuscript this paper is deserving of publication.

Introduction:

Neuroglobin (Ngb), one of novel members of the globin superfamily, is expressed predominantly in brain neurons, and appears to modulate hypoxic-ischemic insults. The mechanisms underlying Ngb-mediated neuronal protection are still unclear. Thus, it would be nice to have longer introduction about Ngb and about Ngb-OCs.

Behavioural test:

1.       Were OFT and EPM performed on the same groups of animals?

2.       Whether these tests were carried out on the same day

3.       Was there any time interval between the performed tests?

Results:

p.248, line 236-237 Authors wrote about the S6 group whereas in the Fig. 3b-c we can observe statistical differences between S2 groups.

Statistical analysis

There is a lack of F values.

Corrections:

-          The lack of the explanations of some abbreviations: MeCP2, HPA

-          Page 3, line 99 – “phenomena”

Author Response

Comments Review #3

The paper describes the in vivo and in vitro testing performed to investigate the role of stress in Ngb expression as a pivotal protein in brain oxygen sensing reactions with expected involvements in anxiety progression. Of particular of interest is the main conclusion of the manuscript - - it seems that the frequency, not the intensity is the key element in the CA3 buffering reactions. Overall this is a well designed study and provides very interesting results. With some minor editing to the manuscript this paper is deserving of publication.

Introduction:

  1. Neuroglobin (Ngb), one of novel members of the globin superfamily, is expressed predominantly in brain neurons, and appears to modulate hypoxic-ischemic insults. The mechanisms underlying Ngb-mediated neuronal protection are still unclear. Thus, it would be nice to have longer introduction about Ngb and about Ngb-OCs.

A: Issue solved. Thank you for you attention. In Introduction, the blue color text highlight the text added to the manuscript.

[Ngb, as well as cytoglobin, in the absence of ligands (eg. oxygen), show hexacoordination by distal histidine which contrasts the pentaccordination hem geometry seen in deoxygenated hemoglobin and myoglobin. Ngb displays high-affinity oxygen binding of approx. 1 Torr and occurs at µM concentrations in the neurons as well as endocrine tissues [34]. However, the clear physiological function of Ngb is still unknown, besides Ngb intervention in hypoxia/stroke reactions that were noticed as enhancers for Ngb expression. The involvement of Ngb in cancer progression was also questionable. This ectopic role of the Ngb was derived from cytoglobin function that was noticed as a cancer progression protein that was over-expressed in different types of tumors (non-small cell lung cancer, esophageal cancer, or melanoma)]

[Few data were reported about Ngb and glial cells. Ngb was noticed as a protein expressed beside neurons by astrocytes and microglia but an expression link between Ngb and oligodendrocytes is still not demonstrated. However, a correlation between OCs and Ngb was noticed in neurodegenerative disorders, autism-spectrum disorders, heavy metals toxicosis, stroke, or traumatic brain injuries [39-41]. The research setup was started in previous studies where Ngb was noticed with different expression patterns [36] in the frontal cortex after long-term treatment with metformin. Also, the methylation of the globins was reported with an ambiguous effect on the expression of the globins [42]. However, DNA methylation in the 5′regions of the globins genes might play a direct role in the regulation of gene expression [43]. Moreover, studies revealed that methylation of the Ngb gene influenced the tissue-specific expression pattern of the protein [44, 45]. Other studies [46, 47] have noticed that MeCP2 was involved in cytoglobin expression and in neuronal plasticity control, but available consistent data about MeCP2 and Ngb relation were not shown. Furthermore, MeCP2-Ngb interaction in stress was also a lack of knowledge that required experimental data.]

Behavioural test:

  1. Were OFT and EPM performed on the same groups of animals?

A: Yes. In Materials and Methods, we described now more exactly all the procedures.

  1. Whether these tests were carried out on the same day

A: Yes, on the same day. We added this detail in the manuscript.

  1. Was there any time interval between the performed tests?

A: Yes. Between OFT and EPM were ~ 4hrs. We added this detail to materials & methods.

[Two different tests were used in our study, such as, OFT and EPM, to assess the general locomotor activity and anxiety of the rodents, on the same groups of animals in the same day, with 4 h between evaluations (OFT at 12 a.m, EPM at 16 p.m.).]

Results:

p.248, line 236-237 Authors wrote about the S6 group whereas in the Fig. 3b-c we can observe statistical differences between S2 groups.

A: Thank you for your attention. Issue solved.

Statistical analysis

There is a lack of F values.

A: The F value for all biochemical analyses was inserted in captions.

Corrections:

-          The lack of the explanations of some abbreviations: MeCP2, HPA

-          Page 3, line 99 – “phenomena”

A: Issue solved. All abbreviations are now fully described.

Round 2

Reviewer 2 Report

The authors took into account all my wishes. In connection with the recommendations of other reviewers, the article has been significantly improved and can be accepted for publication.

Author Response

"The authors took into account all my wishes. In connection with the recommendations of other reviewers, the article has been significantly improved and can be accepted for publication."

Thank you very much indeed for the appreciation and all your valuable suggestions!